# Fabrication, Characterization and Implementation of Thermo Resistive TiCu(N,O) Thin Films in a Polymer Injection Mold

**DOI:** 10.3390/ma13061423

**Published:** 2020-03-20

**Authors:** Eva Oliveira, João Paulo Silva, Jorge Laranjeira, Francisco Macedo, Senentxu Lanceros-Mendez, Filipe Vaz, Armando Ferreira

**Affiliations:** 1Centro de Física, Universidade do Minho, 4710-057 Braga, Portugal; evitao96@gmail.com (E.O.); jpcsjoaopaulo@gmail.com (J.P.S.); fmacedo@fisica.uminho.pt (F.M.); fvaz@fisica.uminho.pt (F.V.); 2Moldit–Indústria de Moldes SA, Rua da Moura, Apartado 28, 3720-903 Loureiro, Portugal; jorge.laranjeira@moldit.pt; 3BCMaterials, Basque Center for Materials, Applications and Nanostructures, UPV/EHU Science Park, 48940 Leioa, Spain; senentxu.lanceros@bcmaterials.net; 4IKERBASQUE, Basque Foundation for Science, 48013 Bilbao, Spain

**Keywords:** injection mold, sensors, temperature coefficient of resistance, thermal properties

## Abstract

This paper presents the development of metallic thermoresistive thin film, providing an innovative solution to dynamically control the temperature during the injection molding process of polymeric parts. The general idea was to tailor the signal response of the nitrogen- and oxygen-doped titanium-copper thin film (TiCu(N,O))-based transducers, in order to optimize their use in temperature sensor devices. The results reveal that the nitrogen or oxygen doping level has an evident effect on the thermoresistive response of TiCu(N,O) films. The temperature coefficient of resistance values reached 2.29 × 10^−2^ °C^−1^, which was almost six times higher than the traditional platinum-based sensors. In order to demonstrate the sensing capabilities of thin films, a proof-of-concept experiment was carried out, integrating the developed TiCu(N,O) films with the best response in an injection steel mold, connected to a data acquisition system. These novel sensor inserts proved to be sensitive to the temperature evolution during the injection process, directly in contact with the polymer melt in the mold, demonstrating their possible use in real operation devices where temperature profiles are a major parameter, such as the injection molding process of polymeric parts.

## 1. Introduction

There is an increasing effort to develop new materials to be used in injecting parts—from noncritical products to very challenging technical parts for the automobile industry—with biocompatible fibers [1,2,3]. For the production of these reinforced plastic parts during an injection molding process, an increase in mold wear occurs, and consequently leads to a shorter lifetime, which significantly increases the process costs, leading to the need to know and tune the entire production process (filling, packing and cooling) to the smallest detail [4,5].

One of the critical parameters of a mold is the temperature control system [6]. This system is the main driver of cycle time; the more efficient, the shorter the time required to inject a part. However, its impact on the quality of the molding, through the control of the cooling, is still relatively high [7]. Its sizing is nowadays empirical, and with negative consequences such as longer cycle times, poor quality due to distortions, bends, flow lines or visual defects, among others. For the effective control of the injection process, it is necessary in some cases to raise the temperature of the molding zone so that it receives the polymer at an elevated temperature, and then the temperature control system efficiently draws out the heat to allow as small a time cycle as possible. With traditional technologies, the strategy of controlling and changing mold temperature is not possible, as it is both expensive and slow.

To solve problems of temperature monitoring during the injection process, it is imperative to include temperature sensors on the molding surface. For that, a resource of resistance temperature detectors (RTDs) is a requisite. In theory, any conductor could be used for an RTD. Nickel (Ni) and platinum (Pt) are commonly used for metal-based transducers, as they do not oxidize at higher temperatures (>300 °C) and show highly linear electrical resistance vs temperature behavior [8]. Another metal that is commonly used in the fabrication of RTDs is copper (Cu), due to its cost benefit. Furthermore, it is fairly linear and has a precision of about 0.1 °C at temperatures below 300 °C [9]. Ni and its alloys are also rather low-cost, with low electrical conductivity and high values of a temperature coefficient of resistance (TCR). Pt is the most-used element to produce RTDs, due to the fact that it is relatively non-reactive, has a well-established TCR and is very accurate in the temperature range of −260–1000 °C [10]. Currently, modern RTDs are usually combinations of oxides such as Ni-O, Mg-O or Cu-O [8]. As an alternative to these oxides, different thin-film systems based on titanium (Ti) are being used in many fields, due to their excellent mechanical properties and good stability at high temperatures [11].

The common method to produce thin films by physical vapor deposition (PVD) uses normal incidence, and consequently columnar films grow normal to the substrate [12]. Thin films produced at oblique angles provide strong benefits compared to conventional sputtering techniques [12,13,14,15]. Indeed, an extensive variety of morphologies can be produced by applying a tilted incidence flux and substrate rotation. Taking this into account, the present work explores two main innovations. First, a new thin film is presented, based on the optimization of a nanostructured zigzag-like architecture, where a nitrogen- and oxygen-doped titanium-copper thin film system (TiCu(N,O)) is prepared and optimized with different doping levels of nitrogen and oxygen reactive gases, in order to obtain the best sensor response in an injection mold. Second, the paper describes the integration of the developed temperature sensor on the molding surface. For this purpose, a set of sensors have been produced directly on steel inserts and integrated in the mold cavity, with the goal of measuring temperature variations of between 35 °C < T < 100 °C during the injection process by the electrical resistance variations.

## 2. Materials and Methods

### 2.1. Thin Film Preparation

Titanium–copper (TiCu) thin films with a concentration of roughly Ti_50_Cu_50_ at % were DC sputtered from a bulk titanium target (with size of 200 mm length, 100 mm width and 6 mm thickness, and 99.96 at % purity), and placed at 70 mm from the substrate using a customized vacuum chamber. Taking into account previous results [16], the Ti target was adapted with constant amounts of Cu pellets (with specific area of ~0.2 cm^2^), symmetrically distributed along the preferential erosion area (~50 cm^2^) [17], as seen in Figure 1a, with the purpose to tune the Cu content in the thin films to around 50 at %, as described in [18].

The angle of the Ti particle flux (α) was measured from the substrate normal, by tilting the substrate holder at 80° [16]. The TiCu(N,O) films were deposited for 20 min, with 5 min of each chevron producing a zigzag-like structure, as seen in Figure 1b. For that, a constant argon flow rate of 25 sccm was applied, and the work gas atmosphere was composed of argon and a mixed reactive N_2_ + O_2_ gas mixture (17:3 ratio). The N_2_ + O_2_ gas mixture flow rate varied from 2 to 10 sccm, corresponding to a variation in the partial pressure from 2.1 × 10^−2^ to 1.3 × 10^−1^ Pa. A reactor from Diener electronic, model Zepto, was used to clean the substrates (glass ISO norm 8037−1 microscope slides and (100) p-type silicon wafers) (Sigma-Aldrich, St. Louis, MO, EUA), using a pure Ar atmosphere at a RF power of 100 W for 900 s.

Before the electrical resistivity vs temperature measurements, an annealing protocol was realized after deposition in order to obtain a stable thermoresistive response. The protocol involves an in-air annealing process of the films, from room temperature to a maximum temperature of 250 °C, with a constant rate of 20 °C/min. This maximum temperature was then held for 60 min, followed by a free cooling stage to room temperature again. At this annealing temperature, thin films suffered several structural and microstructural changes, such as grain growth, recrystallization and structural refinement, among others [19].

### 2.2. Morphological and Structural Characterization

The morphology of the TiCu(O,N) films was characterized by Scanning Electron Microscopy (SEM), using a NanoSEM–FEI Nova 200 (FEG/SEM) microscope (Center for Electron Nanoscopy, Lyngby, Denmark), on top view conditions with a fractured cross-section. The structure of the films was characterized by X-ray diffraction (XRD) (Bruker, Billerica, MA, EUA) using a Bruker D8 discover diffractometer (Cu λ_Kα1_ =1.54060 Å), operating in a θ/2θ configuration and a step of 0.02° per 0.2 s from 30 to 72°.

### 2.3. Electrical Properties

The electrical resistivity of the samples was measured by the 2-wire method. For this, two parallel rectangular gold electrodes (with size of 6 mm length and 2 mm width separated by 1 mm) were deposited on each sample by magnetron sputtering. Copper wires were attached to the electrodes with silver paint to ensure good electrical contact. The voltage applied was between −10 V and 10 V, and the current measured were achieved by a Keithley 487 picoammeter/voltage source (Keithley, Cleveland, Ohio, EUA). All measurements were performed in direct current (DC) mode and at room temperature. The sheet resistivity *ρ* (Ω.sq) was calculated by
(1)$$\rho =R\frac{A}{d}$$
where *R* is the surface resistance, *A* is the electrode area (~19.6 mm^2^) and *d* is the distance between the electrodes (1 mm). The electrical conductivity (*ρ*) is given by the inverse of the electrical resistivity, σ = 1/*ρ.* The measurements were carried out in air, using a custom-made furnace (in-dark conditions and constant humidity). The error associated with the resistivity measurements was below 1%.

### 2.4. Thermoresistive Properties

The relationship of resistance for a resistive temperature sensor with the temperature can be approximated by
(2)$$R_{T}=A∙exp(\beta /T)$$
where *R_T_* is the resistance at temperature *T* (°C), *A* (Ω) and *β* (°C) are constants for the particular thermistor under consideration, and *T* (°C) is the absolute temperature.

The temperature coefficient of resistance is given by:(3)$$TCR=\frac{\Delta R}{R_{0}}∙\frac{1}{\Delta T}=\frac{\beta }{T^{2}}$$

In Equation (3), Δ*R* (Ω) is the variation of the film’s resistance, *R*_0_ (Ω) is the initial film’s resistance at 25 °C and Δ*T* (°C) is the variation of temperature. Equations (2) and (3) are only valid over small temperature ranges, where the slope of the Ln(R_T_) versus 1/T^2^ approximates to a linear relationship. In this work, the TCR values were obtained from the electrical resistance vs temperature, measured in two cycles over the range of 35 to 200 °C, regulated with a Linkam’s LTS420 stage at a rate of 10 °C/min. Typically, industrial platinum resistance temperature detectors have a nominal TCR value of 3.85 × 10^−3^ °C^−1^ [20].

### 2.5. Thermal Properties

Thermal properties were used to analyze the influence of the zigzag nanostructure, together with the N_2_ + O_2_ concentration, on the thermal response of TiCu(O,N)-based films. For that purpose, a nonstationary photothermal technique, modulated infrared photothermal radiometry (MIRR) [21], was used, as seen in Figure 2.

Essentially, the film is heated with a modulated laser beam, and the infrared response at the same frequency is recorded by synchronous detection. The photothermal setup uses a DPSS 532 nm laser for excitation and an acoustic–optic modulator to modulate the incident light from nearly DC up to 100 kHz [22]. The penetration depth of the generated “thermal waves” depend inversely on the modulation frequency, the modulated IR phase signals measured for the thin film–substrate system, information on the thermal diffusion time (τ_s_ = d_s_^2^/α_s_) of the thin film and the ratio of thermal effusivities (e_s_/e_b_) at the thin film–substrate interface, which can be obtained from the inverse calibrated modulated IR phases. The two-layer model and the extremum method are used for the direct quantitative interpretation of the measured data [23,24]. When the thin film thickness (d_s_) is known by a separate measurement, such as SEM, the thermal diffusivity of the thin film (α_s_) can be determined, which characterizes the time-dependent heat distribution in the film, and when the thermal effusivity of the substrate (e_b_) is known by separate measurement, the effusivity (e_s_) of the thin film can be determined. The thermal effusivity (e = √k*ρ*c) controls the transient surface heating processes, the heat transition at the interface between different materials, and is of fundamental importance for the heat propagation across layer systems [19]. Once the thermal diffusivity and effusivity are known, the thermal conductivity k_s_ = e_s_√α_s_ and volume heat capacity (*ρ*c)_s_ = e_s_/√α_s_ of the thin film can be calculated, with ρ being the mass density and *c* the specific heat capacity [24].

### 2.6. Mold Integration

As seen in Figure 3a,b, a set of sensors were produced directly on steel inserts using lift-off photoprocessing [25], and integrated in the mold cavity, as seen in Figure 3c, with the goal to control the temperature during the injection process. The electrical resistance was measured using an Agilent 34401A multimeter (Agilent, Santa Clara, California, EUA) connected to a computer with a LabVIEW software interface, at a rate of 20 Hz [26]. Before that, an insulator layer (TiO_2_) was deposited on the steel insert in order to electrically isolate the thermoresistive layer. This layer was produced with a thickness of around 1 μm. The reactive work atmosphere was a mixture of argon and oxygen, with pressures of 0.28 Pa and of 0.42 Pa, respectively. In order to avoid any influence of the TiO_2_ layer, the same experimental conditions to deposit the TiO_2_ layer were kept unchanged, and thus all the deposition parameters were fixed. In this way, it was assured that all variations in the sensors’ response and different properties resulted from the functional layer itself and not the insulating TiO_2_ layer, which was kept absolutely constant in terms of preparation conditions.

## 3. Results and Discussion

### 3.1. Morphological and Structural Characterization

A set of TiCu(N,O) thin films was sputtered with increasing the N_2_ + O_2_ contents (increasing the reactive flux of the N_2_ + O_2_ mixed gas flow from 0 to 10 sccm, with a fixed incident angle α = 80° to obtain zigzag-like nanostructures. The SEM micrographs of the cross-section and surface are presented in Figure 4.

Figure 4 shows that the zigzag-like morphological changes with increasing N_2_ + O_2_ content. The films reveal an average thickness of about 200 nm. Clear zigzags chevrons are observed in the TiCu thin film prepared without N_2_ + O_2_ flow (Figure 4a_1_), and this behavior is maintained until the samples are prepared with a flux of 8 sccm of N_2_ + O_2_ (Figure 4a_1_ to e_1_). Furthermore, increasing N_2_ + O_2_ flow leads to an increase of the porosity of the films until the sample prepared with a 4 sccm, which shows the wider column gaps (Figure 4c_2_). By further increasing the N_2_ + O_2_ flow to 10 sccm, the zigzag-like structure disappears, the films seem to grow denser than the ones prepared with lower flows, and no columnar gaps are easily detected on the films’ surface.

Moreover, when the N_2_ + O_2_ flow is increased from 0 to 10 sccm, the poisoning level of the Ti target increased. The poisoning level of the target strongly effects the ability of releasing material from the Ti target surface. The relationship between N_2_ + O_2_ flow and deposition rate in poisoning mode is nearly linear, due the linear shape of the target potential, as seen in Figure 4. Consequently, the rise of reactive gas flow in poisoning mode leads to the decreasing of the deposition rate of the TiCu(N,O) coating, as seen in Figure 5.

With the aim to evaluate if the morphological changes correlate with changes in the structure of the films, XRD measurements were made. Figure 6 shows the XRD patterns of the sputtered TiCu thin films deposited on (1 0 0) silicon substrates. For the TiCu film (0 sccm), a larger diffraction peak situated at 2θ–42.5° is observed, with several indexing possibilities due to their similar angular positions: Ti_2_Cu (110) (ICSD–62-9404); TiCu (012) (ICSD–62-9389); Ti_3_Cu_4_ (017) (ICSD–62-9390); Ti_2_Cu_3_ (015) (ICSD–62-9380); and TiCu_2_ (020) (ICSD–62-9379). For this specific diffraction peak, the phase diagram shows that the formation of Ti_3_Cu_4_ and Ti_2_Cu_3_ intermetallic phases are thermodynamically more favorable [16]. By increasing the amount of N_2_ + O_2_ gas flux in the system, the TiN phase appears with a preferential orientation along the (111) (ICDS–38-1420) direction, located at 2θ–36.6°, and followed with steep decreases in the crystallinity degree. Increasing even further the amount of reactive N_2_ + O_2_ gas flow, the XRD peak positions confirm that samples deposited at various N_2_ + O_2_ pressures are all TiN phases [27]. Essentially, a Ti crystal is supposed to be an ‘interstitial’ crystal, where N atoms fit into the gaps in the Ti structure. The N atoms occupy octahedral sites of the Ti lattice as the amount of nitrogen is increased. The Ti lattice is able to accept small amounts of nitrogen at octahedral sites [28]. The results reveal a strong dependence of the film texture on the nitrogen content, following closely the microstructural changes analyzed previously (Figure 4).

### 3.2. Electrical Resistivity

Figure 7a shows that the electrical resistivity increases with increasing the N_2_ + O_2_ content in the samples, which correlate with the previously analyzed variations of the morphological and structural features (Figure 4 and Figure 6).

Taking into account the amount of N_2_ + O_2_ flow, the samples without reactive gas mixture flow exhibited lower resistivity values (1.07 × 10^2^ Ω.sq). For the samples with fluxes in the range of 2 to 10 sccm, the electrical resistivity increases up to 3.00 × 10^7^ Ω.sq, and is kept almost constant until the flux of 8 sccm. For the sample with 10 sccm, the electrical resistivity increases again to 1.17 × 10^8^ Ω.sq.

As discussed previously, the XRD peak positions confirm that samples are all TiN phases, and that there is a strong dependence of the film texture on the nitrogen content. The initial rapid increase of the electrical resistivity for increasing N_2_ + O_2_ concentration up to 10 sccm is attributed to the increase of impurity defects: for the range between 2 and 8 sccm, the N atoms occupy interstitial positions in the hexagonal Ti structure, as supported by the presence of the TiN diffraction peaks (Figure 6). Morphology characterization revealed that the zigzag structure disappears, and that the films seem to grow denser than the ones prepared with lower fluxes, and no gaps are present on the surface with an increasing N_2_ + O_2_ content. The fact that resistivity increases for N_2_ + O_2_ fluxes up to 8 sccm and still increases for flux with 10 sccm proved that morphology variations are not as relevant a parameter as the upper mentioned structural variations for the determination of the films’ electrical of the films.

As the temperature increases from 35 to 200 °C, the electrical resistivity of the TiCu(O,N) thin films decreases, as presented in Figure 7b. This behavior is characteristic of negative TCR (NTCR) thermistors, for which the resistance decreases with increasing temperature. Taking this into account, and as described above, the slope of the ln(R) versus 1/T^2^ (Equation (2)) approximates a linear relationship, as seen in Figure 7b. The values for the NTCR were obtained using Equation (2) and are represented in Table 1.

From Table 1, and taking into account the amount of N_2_ + O_2_ flux, the results of the NTCR follows the same trend found for electrical resistivity. The samples without N_2_ + O_2_ exhibited lower NTCRs values (7.62 × 10^−4^ °C^−1^), which are lower than the ones reported for the Pt detectors (3.85 × 10^−3^ °C^−1^) [20]. In the same way, for samples with fluxes in the range of 2 to 10 sccm, the NTCR increases up to 2.29 × 10^−2^ °C^−1^, for fluxes of 2 sccm, which are ~6 times higher than the Pt detectors, and kept almost constant until the flux of 8 sccm. Finally, for the sample with 10 sccm, the NTCR increases to 2.13 × 10^−2^ °C^−1^. However, Figure 4 shows that for the sample with 10 sccm, the zigzag structure disappears, the films look to grow denser than the ones prepared with lower fluxes, and no gaps are present on the surface. Yet, the results obtained in [29] reveal denser films that show a very poor resistance to stretching, with the electrical resistance increasing sharply after the application of deformation, which demonstrates its inadequacy for the targeted sensor applications in injection molds, as the molds are subjected to compressive forces.

### 3.3. Thermal Proprieties of TiCu(O,N) Thin Films

With the purpose to study the influence of the microstructural evolution on the thermal properties of the TiCu(O,N) thin films, the samples were analyzed by MIRR, and the results are presented on Figure 8 and Table 2.

For lower frequencies, and due to the penetration depth dependence on the modulation frequency, the weak adjustment between the experimental data and the simulation is related with substrate contributions to the measured signal. Furthermore, as the modulation frequency increases, the contribution of the TiCu(O,N) thin film to the measured signal becomes more important, and thus the theoretical curve will adjust better to the experimental data; while for lower frequencies, the influence of the substrate will become dominant.

The observed variations follow the same behavior of the previous results. Increasing the amount of N_2_ + O_2_ flux, the thermal diffusivity of the films decreases from 1.46 × 10^−7^ to 3.07 × 10^−8^ (m^2^/s), in opposition to the NTCR, which increases with an increasing N_2_ + O_2_ flux. The higher values for the thermal diffusivity occur for the sample without N_2_ + O_2_ flux, which also shows the lower value of NTCR. This is certainly related with the intermetallic phase revealed in Figure 6, reflecting a higher internal order and therefore better diffusivity.

In addition to thermal diffusivity, the absolute values found for this parameter are quite low, indicating that the produced films are indeed very poor thermal conductors.

### 3.4. Transducers Integration and Injection Tests

Taking into account the previous results, the TiCu(O,N) thin films prepared with a reactive gas mixture flow of 2 sccm were placed in steel inserts, as shown in Figure 9. In this figure, it is possible to observe the thin film system integrated on the surface of the injection mold, the thermoresistive transducer, the electrode structures, the top and basecoat (which looks like a single layer), and finally the serial USB port to connect to the PC and acquire the signal.

In order to validate the temperature measurements, an experimental protocol was planned to evaluate the viability of the transducers, as well as to determine possible problems that could be associated with the mounting of the transducers and the collection of data.

Figure 10 shows representative measurements of the integrated steel insert with the thermoresistive transducer before the injection cycle, and during repeated heating and cooling cycles from 40 °C to 70 °C with a custom-made heater. The cooling was controlled with a Piovan TMW regulator equipment. It is observed that the response of the system is the same, as demonstrated previously in Figure 7. The measurement results in Figure 10 show that the temperature increase causes a steep decrease of the electrical resistance in every position, increasing when the water cooling is on. The sensor structures are thus sensitive enough to be used as an indicator for the quality of the produced part. To compare the results of the sensors tested on glass with the results presented in Figure 10, one needs to understand the dynamics of the heat transfer in steel promoted by a fluid that could be modeled with a finite element model, and quantify the influence of the steel on the measurement results. The most challenging part in the modelling is making the right choice of the values of the heat transfer coefficient and the heat distribution coefficient. The heat distribution coefficient indicates how much thermal energy flows into the steel piece. The heat transfer coefficient indicates how much thermal energy is lost from the steel piece due to the use of cooling lubricant. As of this moment, however, this is not the aim of this work. Therefore, after the stabilization of the mold temperature at 40 °C, a set of polymer injection procedures have been carried out with polypropylene as an injection material, and the electrical resistance variation was measured, as shown in Figure 10.

Figure 11a,b shows the results of three injection cycles, demonstrating that when the temperature of the mold increases to ~100 °C, a decrease in the electrical resistance occurs, which is followed by a decrease in the resistance caused by the injection of the polymer. However, a reduction in electrical resistance was expected to be larger than that observed due to the injection temperature of the polymer at around 200 °C, as seen in Figure 11b. In our case, the sensors are placed at the ends of the mold cavity (Figure 3c), and the melt polymer is forced into a cold empty cavity with a desired shape, and is then allowed to solidify under a high holding pressure. The lower decrease in the electrical resistivity is related to the distance between the point of the polymer injection and the sensor location, as well as the rapid solidification of the polymer [30,31,32,33]. In this sense, the system can be improved by the integration of the thin film sensor systems directly in contact with the polymer melt in the mold.

Overall, the present work successfully demonstrates that the measurement of the temperature distribution in the mold directly during the heating and cooling process is possible, allowing improvements in the life of the mold, and consequently, the quality of the injection parts, and also that the TiCu(O,N) system is suitable for the development of the required thermoresistive sensors.

## 4. Conclusions

The present work reports on the development of steel inserts with metallic thermoresistive TiCu(O,N) thin films to evaluate an innovative solution to dynamically control the temperature of the injection molding process of polymeric parts. The TiCu(O,N) systems can be optimized depending on the flux of N_2_ + O_2_, the morphological, structural and low thermal diffusivity variations leading to a thermoresistive response of up to 2.29 × 10^−2^ °C^−1^, which are ~6 times higher than the platinum (Pt) detectors (3.85 × 10^−3^ °C^−1^).

The integration of the developed thermoresistive films in the injection molding system will allow the comprehensive evaluation of the process parameters (especially those affecting the temperature of the polymer during the injection molding cycle), which assumes a relevant importance in the product quality and lifetime of the mold, and will support technical decisions in relation to the potential defects and modifications on the mold, when preventive adjustments are required.

## Figures and Tables

**Figure 1 materials-13-01423-f001:**
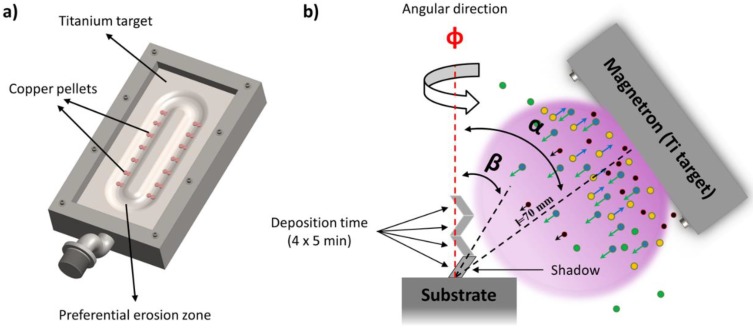
(**a**) Representation of the distribution of the Cu pellets on the Ti target; (**b**) α is the applied angle of the substrate relative to the Ti particle flux, β is the chevron growth angle, Φ is the angular direction and l = 70 mm is the distance between the target and the substrate.

**Figure 2 materials-13-01423-f002:**
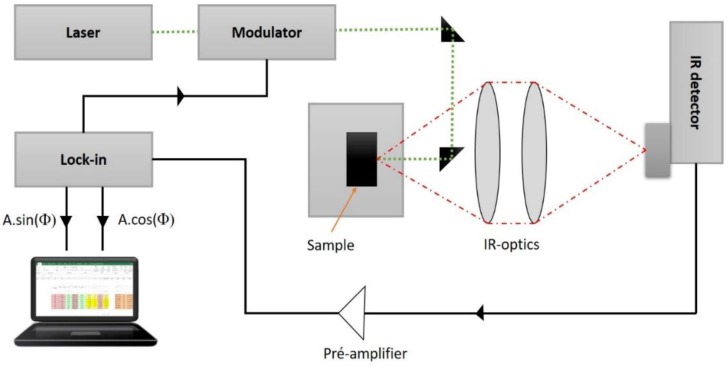
Schematic representation of the photothermal measurement device, comprising a laser (532 nm), an acoustic-optical modulator, Infra-Red (IR) optics and an IR detector, as well as a computer-controlled lock-in amplifier.

**Figure 3 materials-13-01423-f003:**
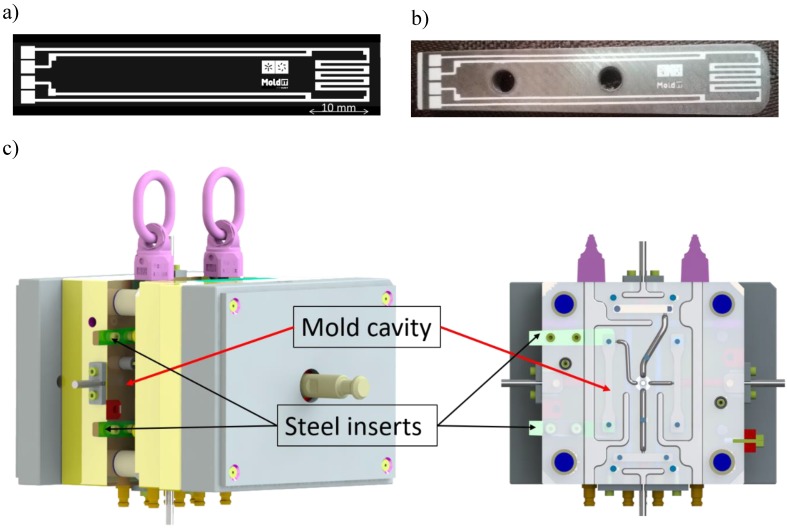
(**a**) Draw of the designed sensor; (**b**) draw transferred to the steel inserts by lift-off; and (**c**) schematic representation of the mold cavity and the place where the steel inserts are placed.

**Figure 4 materials-13-01423-f004:**
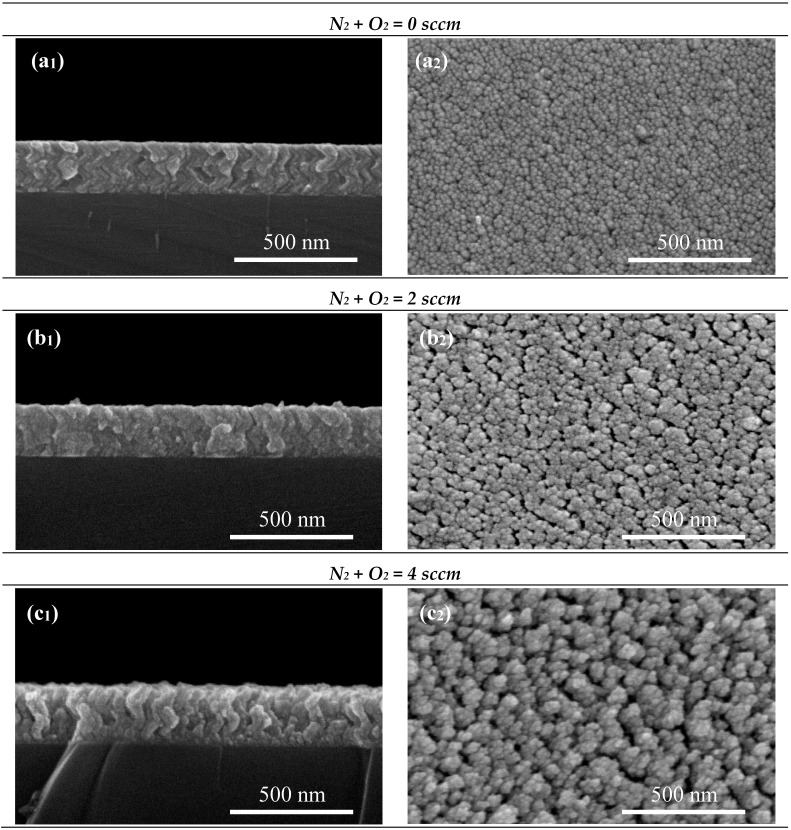
SEM of the fractured cross-section views (**a****_1_**–**f****_1_**) and surface (**a****_2_**–**f****_2_**) of the TiCu(O,N) thin films deposited on silicon sputtered with different concentrations of N_2_ + O_2_. (**a_1_**–**a_2_**) (N_2_ + O_2_) flow = 0 sccm, (**b_1_**–**b_2_**) (N_2_ + O_2_) flow = 2 sccm, (**c_1_**–**c_2_**) (N_2_ + O_2_) flow = 4 sccm, (**d_1_**–**d_2_**) (N_2_ + O_2_) flow = 6 sccm, (**e_1_**–**e_2_**) (N_2_ + O_2_) flow = 8 sccm, (**f_1_**–**f_2_**) (N_2_ + O_2_) flow = 10 sccm.

**Figure 5 materials-13-01423-f005:**
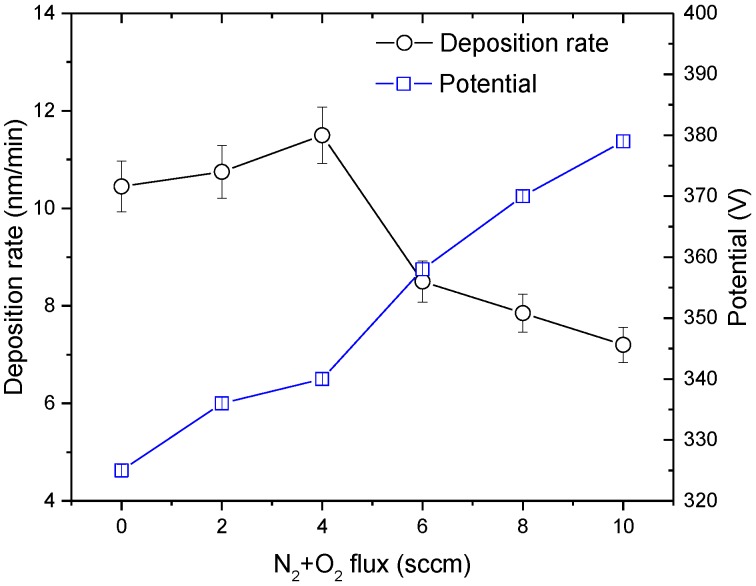
Deposition rate of TiCu(O,N) thin films and target potential as a function of the increasing N_2_ + O_2_ flux.

**Figure 6 materials-13-01423-f006:**
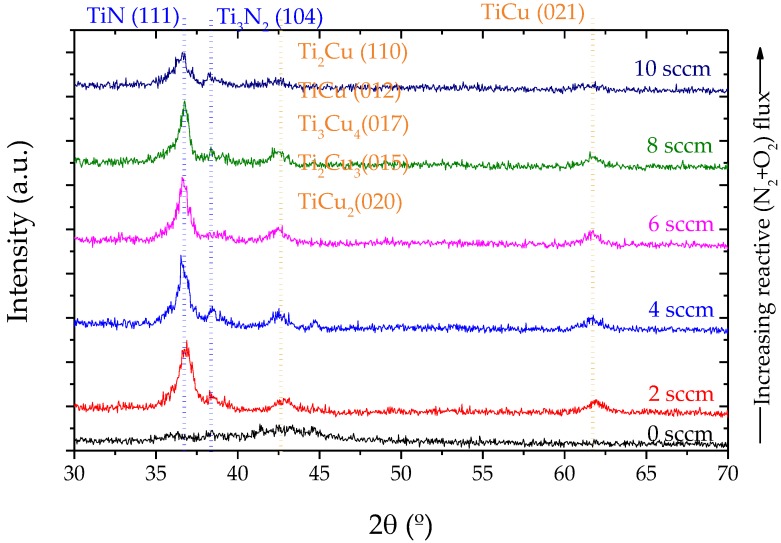
XRD diffractograms of the nanostructured TiCu(O,N) thin films.

**Figure 7 materials-13-01423-f007:**
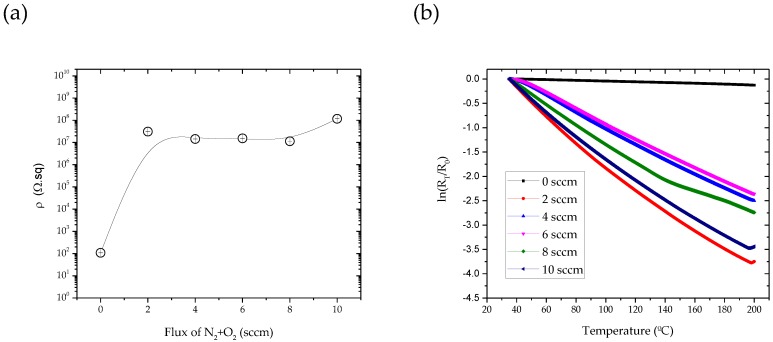
(**a**) Room temperature electrical resistivity as a function of the N_2_ + O_2_ flux and (**b**) thermoresistive response of samples of TiCu(O,N) thin films sputtered on glass substrates.

**Figure 8 materials-13-01423-f008:**
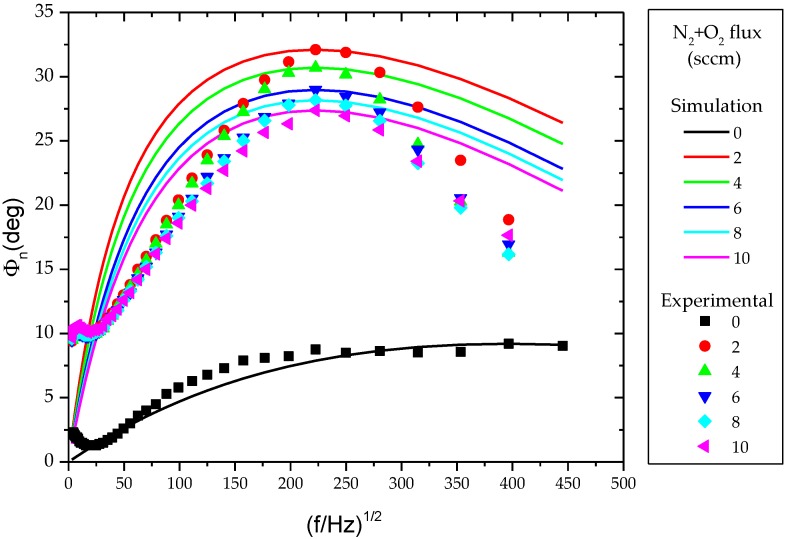
Inverse normalized phase vs. modulation frequency for the N_2_ + O_2_ flux. The solid lines represent the simulations based on the two-layer model proposed by Fotsing et al. [24].

**Figure 9 materials-13-01423-f009:**
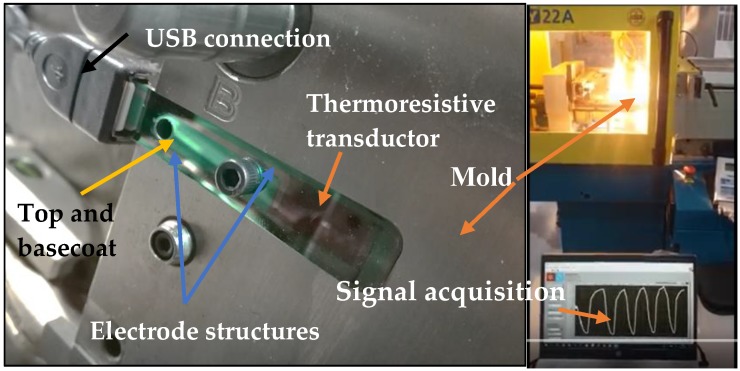
Integrated steel insert with the different layers, top-coat, base-coat, thermoresistive transducer and electrodes, placed in the mold surface.

**Figure 10 materials-13-01423-f010:**
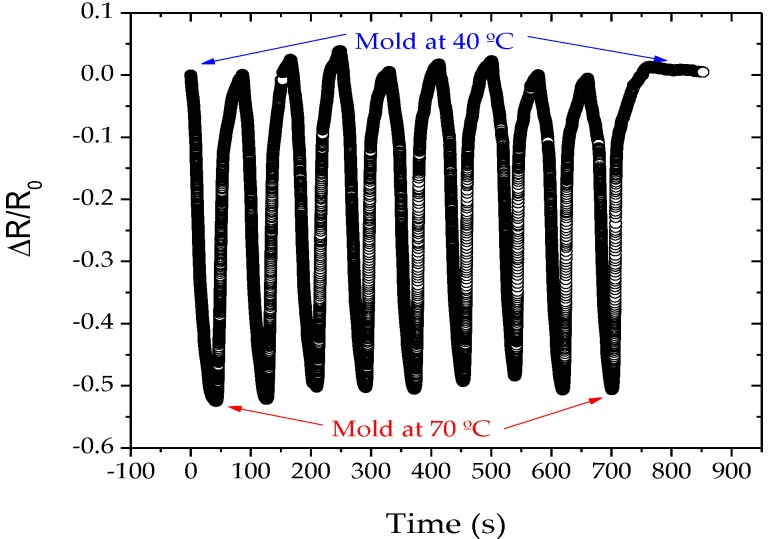
Electrical response of the integrated TiCu(O,N) thermoresistive sensor prepared with a flux of 2 sccm of N_2_ + O_2_ during repeated temperature cycles.

**Figure 11 materials-13-01423-f011:**
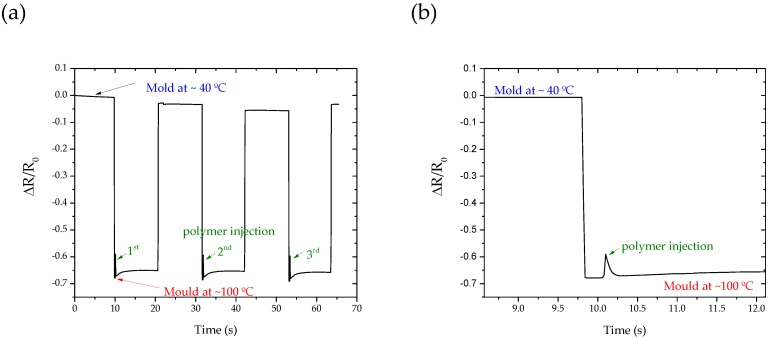
Electrical behavior of the integrated TiCu(O,N) system prepared with a flux of 2 sccm of N_2_ + O_2_ during successive injection cycles. (**a**): The first three injection cycles and in (**b**) a zoon of the 1st injection cycle.

**Table 1 materials-13-01423-t001:** Values of NTCR for the TiCu(O,N) thin films sputtered on glass substrates.

Flux of N_2_ + O_2_ (sccm)	NTCR °C^−1^	Error
*0*	7.62 × 10^−4^	1.24 × 10^−6^
*2*	2.29 × 10^−2^	6.52 × 10^−5^
*4*	1.59 × 10^−2^	2.01 × 10^−5^
*6*	1.51 × 10^−2^	1.59 × 10^−5^
*8*	1.68 × 10^−2^	7.01 × 10^−5^
*10*	2.13 × 10^−2^	5.45 × 10^−5^

**Table 2 materials-13-01423-t002:** Thermal parameters of the samples determined by MIRR, from the data in Figure 8. Thermal diffusivity (τ_s_ = d_s_^2^/α_s_) values were calculated assuming the thickness (*d_s_*) estimated by SEM.

Flux of N_2_ + O_2_ (sccm)	d_s_ (μm)	(*f*/Hz)^−1/2^	τ_s_ (s) × 10^−7^	*e* _s_ */e* _b_	α_s_ (m^2^/s) × 10^−8^
0	0.21	396,59	3.02	1.66	14.59
2	0.22	222,58	5.46	9.61	8.52
4	0.23	222,58	5.88	8.17	9.13
6	0.17	222,58	6.37	6.81	4.57
8	0.16	222,58	6.58	6.29	3.77
10	0.14	222,58	6.78	5.84	3.07

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
