# Peer review of "Fabrication, Characterization and Implementation of Thermo Resistive TiCu(N,O) Thin Films in a Polymer Injection Mold"

_materials, 2020, doi:10.3390/ma13061423_

Round 1

Reviewer 1 Report

The work contains interesing results in terms of fabriation of thin films using GLAD method and controlling the crystal growth and thin film resistance. The data show better TCR values, although it does not convince be me fully that the developed material without discussion of stability will be better for application that standarized Pt sensors. Before being published few major points are misssing, (apart few typos or styling errors, which I listed at the end)

In article I haven't found the discussion about 

1) Amount of Cu content in each film. Do you have any results on Cu concentration? If not I suggest to measure it quantitatively using EDX or XPS. 

2) What about the reproducibility of the fabrication process? 

3) Thermal stability of this material, degradation time.

Styling and typing errors I have found in lines:

21) -2 and -1 should be put in the power

32) First The should be removed

138) Equation 3 (I am mathemathician I like to use "x" only for the vector product, and · for mulitplication)

185) Thickness of ... m?

237) Missing units

238) Missing units

239) Figure 6 b) Lets do the lines and marks in the same color. Logarithm in the caption"ln" as in line 257)

Reviewer 2 Report

In the study by Oliveira et al., the authors have described how they developed and tested metallic thermoresistive thin films used to control dynamically the temperature during the injection molding process. The results are interesting but I have a few suggestions and questions:

1. Could you provide more information on resistance measurement using the 2-wire method (119-122)? (voltage, current, gold electrodes and sample size)

2. Before producing sensors on steel inserts, the authors deposited TiO2 layer. The article does not provide information on the impact of this layer on the formation of the TiCu(N,O) layer and especially on the electrical and thermal properties of the sensor. (185 – incorrect unit?)

3. The electrical resistance of set sensors produced directly on steel inserts was measured using Agilent 34401A multimeter. The presented sensor geometry suggests that the 4-wire method was used. What was the sample rate of resistance measurement for the results presented in Fig.9 and Fig.10?

4. Fig.8 – wrong figure caption - piezo- and thermo-resistive transducer?? (309-310)

5. Fig. 6 showed the thermoresistive response of samples (TiCu-O-N - deposited on glass substrates). What was the measurement procedure? What type of the reference temperature sensor and the thermostatic chamber have been used? Why the authors did not study the sensor on a steel insert with this procedure? The sensors on steel inserts are used in the final injection system and these results are necessary to proove the correct operation of these sensors.

6. The authors presented the electrical response of the integrated TiCu-O-N thermoresistive sensor during repeated heating and cooling cycles (Fig.9). In presented form it is not possible to observe that the response of the system is the same as demonstrated previously in Fig.6 (samples deposited on glass substrate). This requires the presentation of all data on a common plot.

7. Why did not the authors present the transient response of the sensors on steel inserts for heating and cooling to compare the dynamic (additional plot)? Is thermal hysteresis observed?

8. All test descriptions for electrical properties should be extended and supplemented.

9. Some editorial errors noticed: 32 - “The There is…”; 123, 134, 138 – format of formulas (use interchangeably · and x); 343 – “… by by …”.

Reviewer 3 Report

In this paper, the authors used sputtering method to deposit thin films of TiCu(N,O) films and optimize the reactive gas flow rates to achieve high TCR values, and demonstrated proof-of-concept thermoresistive sensing capabilities of the films. The paper is presented with both experimental and modeling results and clearly presented. A few minor comments are:

  1. "control dynamically" should be "dynamically control" in the abstract.
  2. In the abstract, "thermos resistive" should be "thermoresistive".
  3. The values units for the TCR value should be correctly supercripted in the abstract.
  4. "The" in the first line of the introduction should be removed.
  5. Missing symbols across the text especially β, μ and Ω.
  6. How far is the target from the substrate? Please add the distance either in the text or Figure 1.
  7. What is the crystal orientation of the Si wafer? The crystal orientation would strongly influence the growth of the TiCu(N,O) layer.
  8. Explanations for the impact of high N2 and O2 should also discuss the effect of target poisoning, which is shown by the reduced deposition rate in Table 2 and increased resistivity in Figure 6.
  9. Equations are wrongly aligned right.
  10. Missing scale bars in Figure 3.
  11. How were the errors calculated in Table 2?
  12. What would be the approximate maximum temperature that the films can operate reliably?

Round 2

Reviewer 1 Report

I have only one missing point (Point 2):

That I havent found any quantitative data about the thin film composition, neither in this paper, nor in the ref 16. How was the composition of 50 at.% and 50 at. % of Ti measured?

Author Response

The reviewer is absolutely right. The RBS analysis, published by the authors are in the ref (18), we did not detect the reference mistake in the last revision. Spectrometry (RBS) technique was used to measure the atomic composition of the as-deposited samples using a 1.5 or 2 MeV 4He beam, at normal incidence, in a small (RBS) chamber of the IST/LATR 2.5 MeV Van der Graaf accelerator. The composition profiles for the as-deposited samples were determined using the software NDF, after three different measurements for each sample. The resolution in the determination of atomic concentration of each sample was about 3 at.%. The name we suggest in this paper Ti50Cu50 at.% was related with the sample with an amount of 51±3 at.% described in [18].

The text was changed accordingly, line 83 ).

  1. Ferreira, A.; Borges, J.; Lopes, C.; Martin, N.; Lanceros-Mendez, S.; Vaz, F. Piezoresistive response of nano-architectured TixCuy thin films for sensor applications. Sensors Actuators, A Phys. 2016, 247, 105–114.

Reviewer 2 Report

Dear Authors,

thank you for the answers to my questions and consideration of my suggestions. Additionally I suggest to study dynamic parameters of the final sensor set (steel insert) under controlled conditions (thermostatic chamber) and using a reference calibrated temperature sensor.

Author Response

First of all, we would say thank you for your professionalism, kindness and for the positive feedback on our manuscript. Further work will be based on your suggestion, “The study of dynamic parameters of steel insert under controlled conditions”. In this case the aims of this work are a proof of concept approach, where we demonstrate that thin films developed could be used as temperatures sensors and, of course,  we agree with the reviewer, future work should be based on the final set to be implemented in the industry.